# Dabigatran Etexilate Induces Cytotoxicity in Rat Gastric Epithelial Cell Line via Mitochondrial Reactive Oxygen Species Production

**DOI:** 10.3390/cells10102508

**Published:** 2021-09-22

**Authors:** Hiromi Kurokawa, Atsushi Taninaka, Hidemi Shigekawa, Hirofumi Matsui

**Affiliations:** 1Faculty of Medicine, University of Tsukuba, Tsukuba, Ibaraki 305-8571, Japan; hmatsui@md.tsukuba.ac.jp; 2Faculty of Pure and Applied Sciences, University of Tsukuba, Tsukuba, Ibaraki 305-8573, Japan; jun_t@bk.tsukuba.ac.jp (A.T.); hidemi@bk.tsukuba.ac.jp (H.S.)

**Keywords:** dabigatran etexilate, mitochondrial reactive oxygen species, lipid peroxidation, atomic force microscopy, antioxidant

## Abstract

Dabigatran is a novel oral anticoagulant that directly inhibits free and fibrin-bound thrombins and exerts rapid and predictable anticoagulant effects. While the use of this reagent has been associated with an increased risk of gastrointestinal bleeding, the reason why dabigatran use increases gastrointestinal bleeding risk remains unknown. We investigated the cytotoxicity of dabigatran etexilate and tartaric acid, the two primary components of dabigatran. The cytotoxicity of dabigatran etexilate and tartaric acid was measured in a cell viability assay. Intracellular mitochondrial reactive oxygen species (mitROS) production and lipid peroxidation were measured using fluorescence dyes. Cell membrane viscosity was measured using atomic force microscopy. The potential of ascorbic acid as an inhibitor of dabigatran cytotoxicity was also evaluated. The cytotoxicity of dabigatran etexilate was higher than that of tartaric acid. Dabigatran etexilate induced mitROS production and lipid peroxidation and altered the cell membrane viscosity. Ascorbic acid inhibited the cytotoxicity and mitROS production induced by dabigatran etexilate. Therefore, we attributed the cytotoxicity of dabigatran to dabigatran etexilate, and proposed that the cytotoxic effects of dabigatran etexilate are mediated via mitROS production. Additionally, we demonstrated that dabigatran cytotoxicity can be prevented via antioxidant treatment.

## 1. Introduction

Dabigatran and warfarin are anticoagulant drugs used to prevent stroke in patients with atrial fibrillation [1]. Warfarin is a vitamin K antagonist that inhibits the activation of clotting factors II, VII, IX, and X [2]. Vitamin K antagonists exhibit multiple interactions with food and drugs; therefore, patients who are prescribed these drugs need to be monitored frequently. Dabigatran is a novel oral anticoagulant that was approved by the FDA in 2010 [3]. Dabigatran directly inhibits free and fibrin-bound thrombin and exhibits rapid and predictable anticoagulant effects [4,5]. Dabigatran prevents embolic events in patients with non-hemorrhagic stroke and atrial fibrillation and can be used in a safe and effective manner without the need for monitoring and dose adjustment, in contrast to warfarin, which requires continuous monitoring [6]. Moreover, dabigatran is well-tolerated, exhibits predictable pharmacokinetics, and induces effective anticoagulant effects. Therefore, the use of dabigatran is increasing worldwide.

Dabigatran capsules are composed of cores of tartaric acid coated with dabigatran etexilate, which is a prodrug of dabigatran. As a prodrug, dabigatran etexilate does not directly inhibit thrombin. After oral administration, the low pH in the stomach facilitates the conversion of dabigatran etexilate into dabigatran, the active form that acts as a direct thrombin inhibitor [7]. Moreover, the presence of tartaric acid promotes the acidity of the environment, which further assists in the conversion of dabigatran etexilate to dabigatran [8]. The widespread use of dabigatran has led to increased reports of gastrointestinal (GI) bleeding risk associated with its use [9]. Additionally, the risk of GI bleeding associated with dabigatran has been found to be higher than that associated with warfarin [3,10]. Several researchers have reported that tartaric acid plays an important role in the induction of GI bleeding [5,11]. It has been postulated that tartaric acid adheres to the esophagus and damages the esophageal mucosa, which sheds after peristalsis [12]. However, the precise reason why dabigatran use increases the risk of GI bleeding remains unknown.

To investigate why dabigatran use is associated with an increased risk of bleeding, we investigated the role of reactive oxygen species (ROS), especially mitochondrial ROS (mitROS). In a previous report, we investigated the relationship between certain drugs and mitROS. We found that indometacine and bisphosphonate induce cytotoxicity via mitROS production [13,14]. However, it has been found that antioxidants can prevent mitROS cytotoxicity [15,16]. MitROS induce lipid peroxidation and DNA damage [17] and act as mediators of intracellular signaling cascades that can trigger mitochondria-associated damaging events, such as apoptosis via regulation of the Bcl-2/Bax balance [18]. Therefore, any agents, such as antioxidants, that reduce mitROS production can help prevent cellular injury and inhibit dabigatran etexilate-associated cytotoxicity.

In this study, we investigated the cytotoxicity of dabigatran etexilate and tartaric acid, the two main components of dabigatran, using a rat gastric epithelial cell line (RGM-1). We hypothesized that dabigatran etexilate or tartaric acid induces mitROS production, thereby causing cytotoxicity in normal cells and inducing GI bleeding when dabigatran is used. As indometacine and bisphosphonate induce GI bleeding and cytotoxicity via mitROS production, we focused on the mitROS. We also investigated whether antioxidants can attenuate the cytotoxicity of dabigatran etexilate and/or tartaric acid.

## 2. Materials and Methods

### 2.1. Cell Culture

Cells from the rat gastric epithelial cell line RGM-1 were purchased from the Riken Cell Bank (Ibaraki, Japan) and cultured in DMEM/F12 with L-glutamine (Life Technologies Japan Ltd., Tokyo, Japan). The culture medium was supplemented with 10% inactivated fetal bovine serum (GE Healthcare Life Sciences, Amersham Place, UK) and 1% penicillin/streptomycin (Wako Pure Chem. Ind. Ltd., Osaka, Japan). The cells were cultured in 5% CO_2_ at 37 °C.

### 2.2. Cell Viability Assay

The cell viability was evaluated using a Cell Counting Kit-8 (Dojindo, Tokyo, Japan) according to manufacturer’s protocol. Cell Counting Kit-8 is a kit for measuring the number of cells in cell proliferation or chemical sensitivity tests. WST-8 (2-(2-methoxy-4-nitrophenyl)-3-(4-nitrophenyl)-5-(2,4disulfophenyl)-2H-tetrazolium, monosodium salt), which is a reagent, is reduced by intracellular dehydrogenase to produce water-soluble formazan, and the absorbance of this formazan at 450 nm can be directly measured to easily determine the number of viable cells. Cells were seeded on 96-well plates at a density of 2.5 × 10^3^ cells/well and incubated overnight. The supernatant was aspirated and replaced with a medium containing 1 or 5 μM dabigatran etexilate (Chemscene, LLC, Monmouth Junction, NJ, USA) or tartaric acid (Wako). The cells were then incubated at 37 °C for 24 h. After incubation, the cells were incubated with 10% Cell Counting Kit-8 reagent. The absorbance was measured at 450 nm using a Synergy H1 microplate reader (BioTek Instruments Inc., Winooski, VT, USA). The cells were treated with ascorbic acid to determine its effect. Cells were seeded on 96-well plates at a density of 2.5 × 10^3^ cells/well and incubated overnight. The supernatant was aspirated and replaced with a medium containing 0, 50, or 100 μM dabigatran etexilate with or without 500 μM ascorbic acid (Wako). The cells were incubated at 37 °C for 24 h, and cell viability was measured using the Cell Counting Kit-8.

### 2.3. Intracellular mitROS Measurement

MitoSOX (Thermo Fisher Scientific K.K., Kanagawa, Japan) was used to detect mitROS. Cells were seeded on 96-well plates at a density of 1 × 10^4^ cells/well and incubated overnight. The supernatant was aspirated and replaced with a medium containing 0, 10, 15, 20, or 25 μM dabigatran etexilate. The cells were then incubated at 37 °C for 6 h. After aspirating the supernatant, the cells were incubated with 5 μM MitoSOX for 10 min. After incubation, the medium was replaced with MSF buffer, containing 5.4 mM KCl, 136.9 mM NaCl, 8.3 mM glucose, 0.44 mM KH_2_PO_4_, 0.33 Na_2_HPO_4_, 10.1 mM HEPES, 1 mM MgCl_2_.6H_2_O, and 1 mM CaCl_2_.2H_2_O. Fluorescence intensity was measured using a fluorescence microscope (Olympus, IX83). MitoSOX excitation was induced at 535–555 nm, and the emission spectra were recorded using a 570–625 nm filter. To determine the effect of ascorbic acid, the cells were seeded on 24-well plates at a density of 1 × 10^5^ cells/well and incubated overnight. The supernatant was aspirated and the cells were incubated in a medium containing 25 μM dabigatran etexilate with or without 500 μM ascorbic acid. We confirmed that this ascorbic acid concentration did not cause cell injury in RGM1 (data not shown). The cells were incubated for 6 h, the supernatant was aspirated, and MSF was added. The fluorescence intensity of MitoSOX was measured using fluorescence microscopy.

### 2.4. Evaluation of Lipid Peroxidation

Lipid peroxidation was measured using diphenyl-1-pyrenylphosphine (DPPP) (Dojindo). Cells were seeded on a 6-well plate at a density of 5 × 10^4^ cells/well and incubated for 2 days. The supernatant was aspirated, and the cells were incubated in a medium containing 10 µM DPPP for 10 min. After incubation, the supernatant was aspirated, and the cells were incubated in a medium containing 0, 25, 50, and 100 μM dabigatran etexilate. The cells were incubated for 1 h, the supernatant was aspirated, and FluoroBrite DMEM (Thermo) was added. The fluorescence intensity of DPPP was measured using fluorescence microscopy. DPPP fluorescence excitation was induced at 340–390 nm, and the emission spectra were recorded at 420 nm.

### 2.5. Measurement of Cell Membrane Viscosity using Atomic Force Microscopy

The cultured cells were observed using AC mode atomic force microscopy (AFM), and the viscoelastic loss tangent (tan δ) was estimated from the relationship between the measured amplitude image and the phase image [19]. The value of tan δ was estimated with following equation.
$$\tan\delta =\frac{\frac{\omega }{\omega_{\mathrm{free}}}\frac{V}{V_{\mathrm{free}}}-\sin\;\varphi }{Q\frac{V}{V_{\mathrm{free}}}\left(1-\frac{\omega^{2}}{\omega_{\mathrm{free}}^{2}}\right)-\cos\;\varphi }$$

Here, w, Q, V, and j are the cantilever drive frequency, quality factor, the cantilever amplitude, and the phase, respectively. w_free_ and V_free_ are the free reference values of the drive frequency and the amplitude when the cantilever is separated from the sample by a height of Δz.

The viscoelastic loss tangent tan δ of a material is a dimensionless parameter that measures the ratio of energy dissipated to energy stored in one cycle of a periodic deformation [20]; it is the parameter indicating the viscoelasticity of the cell. The cells were seeded in a 60 mm dish. After incubation, the cells were stimulated with 25 µM dabigatran for 6 h. After exposure, the cell membrane viscosity was measured using AFM in the loss tangent mode [19,20].

### 2.6. Measurement of The Mitochondrial Membrane Potential using JC-1

The mitochondrial membrane potential was measured using JC-1 (5,5’,6,6’-tetrachloro-1,1’,3,3’-tetraethyl benzimidazolyl carbocyanine iodide/chloride) [21]. Cells were cultured in a 4-well chamber at 2.5 × 10^4^ cells/well and incubated for 5 days. The supernatant was aspirated and the cells were incubated in a medium with or without 25 μM dabigatran etexilate for 6 h. After incubation, the supernatant was aspirated and the cells were incubated in a medium containing 2 μM JC-1. The cells were then incubated for 30 min. The supernatant was aspirated and the cells were rinsed twice with PBS, following which MSF was added. JC-1 red fluorescence excitation was induced at 535–555 nm, and the emission spectra were recorded at 570–625 nm. JC-1 green fluorescence excitation was induced at 460–480 nm, and the emission spectra were recorded at 495–540 nm.

### 2.7. Statistical Analysis

Each experiment was performed independently at least three times. Data are expressed as the means ± SD and were assessed by an analysis of variance. Individual groups were compared by Tukey’s post hoc or Student’s *t*-test with *p* < 0.05 considered statistically significant.

## 3. Results

### 3.1. Dabigatran Etexilate induces Cytotoxicity in Normal Gastric Cells

The cytotoxicities of both dabigatran etexilate and tartaric acid were analyzed using the WST assay. Cells were stimulated with different concentrations of dabigatran etexilate and tartaric acid for 24 h. The cytotoxicity of dabigatran etexilate increased significantly in a dose-dependent manner (Figure 1A), while tartaric acid did not induce cytotoxicity at concentrations less than 100 μM (Figure 1B). The cytotoxicity of dabigatran etexilate was higher than that of tartaric acid. The IC50 of dabigatran was 26.3 µM.

### 3.2. Dabigatran Increases mitROS Production 

Intracellular mitROS production was measured using MitoSOX. Cells were incubated in a medium with or without dabigatran etexilate. The cell viability in 25 µM dabigatran etexilate was decreased down to 50% for 24 h incubation (Figure 1A). In this study, to evaluate the mechanisms of cell injury by dabigatran etexilate, we selected the 0 to 25 μM concentration and 6 h incubation time. The fluorescence intensity of MitoSOX was measured. The fluorescence microscopic images showed that the fluorescence intensity of MitoSOX increased in a dose-dependent manner (Figure 2A). In particular, the intracellular fluorescence intensity of MitoSOX increased significantly upon treatment with 20 and 25 μM dabigatran etexilate compared to that obtained upon treatment with 0 μM dabigatran etexilate (Figure 2B). 

### 3.3. Dabigatran Etexilate induces Lipid Peroxidation

Since ROS induce lipid peroxidation, we measured the lipid peroxidation of cells using DPPP. The cells were incubated in a medium with or without dabigatran etexilate. After incubation, the fluorescence intensity of DPPP increased in a dose-dependent manner, as evident in the fluorescence microscopic images (Figure 3A). The intracellular fluorescence intensity of DPPP increased significantly upon treatment with 50 and 100 μM dabigatran etexilate compared to that obtained upon treatment with 0 μM dabigatran etexilate (Figure 3B).

### 3.4. Dabigatran Etexilate Alters Cell Membrane Viscosity

Cell membrane viscosity was measured using AFM. Figure 4A,B show the phase-contrast images obtained using an optical microscope. AFM was performed to observe the area surrounded by the dashed line in Figure 4A,B. Figure 4C–J show the AFM images of the topography, amplitude, phase, and loss tangent for dabigatran etexilate (−) and (+) groups, respectively. The AFM amplitude images indicate the elasticity component, and the AFM phase images indicate the viscosity component. The loss tangent was estimated from the amplitude and phase value, which together indicate the viscoelasticity component of the cell. Comparing Figure 4E–H, the stripe images in Figure 4E,G are clearer than those in Figure 4F,H. The loss tangent values of the stripe structure in Figure 4H,J were greater than those of the cell membrane, indicating greater cytosolic elasticity. Therefore, the dabigatran etexilate (−) cells, which more clearly showed the stripe structure (as observed using AFM), were more elastic. The values of tan δ in Figure 4J are lower than those in Figure 4I. The most frequent values of the loss tangent in Figure 4G,H were estimated to be 1.1 and 0.25, respectively (Figure 4K,L). Tan δ is the parameter indicating the viscoelasticity of the cell. Dabigatran etexilate treatment decreased the elasticity and increased the viscosity of the cell. The value of the loss tangent in dabigatran etexilate-treated cells was lower than that in non-treated cells. This suggests that the dabigatran etexilate treatment promoted cell viscosity.

### 3.5. Ascorbic Acid Suppresses the Cytotoxicity of Dabigatran Etexilate via the Inhibition of mitROS Production

The cells were incubated with ascorbic acid to suppress the cytotoxicity of dabigatran etexilate. Compared to untreated cells, the ascorbic acid-treated cells remained viable (Figure 5). Ascorbic acid can inhibit ROS generation induced by dabigatran etexilate, as measured using MitoSOX. Dabigatran etexilate induced intracellular ROS production, which was subsequently inhibited in response to treatment with ascorbic acid (Figure 6A,B). Moreover, the mitochondrial membrane potential was observed using JC-1 under a fluorescence microscope. The ratio of red/green fluorescence in the dabigatran etexilate-treated cells was lower than that in the untreated cells. Moreover, this ratio in the dabigatran etexilate cells treated with ascorbic acid was greater than that in the cells without ascorbic acid, suggesting that dabigatran etexilate-induced decrease in the mitochondrial membrane potential was suppressed by ascorbic acid (Figure 7A,B).

## 4. Discussion

In this study, we investigated the roles of dabigatran etexilate and tartaric acid in order to elucidate the mechanism underlying dabigatran cytotoxicity. We found that dabigatran etexilate induced greater cytotoxicity in normal mucosal cells than tartaric acid via induction of mitROS production. Dabigatran etexilate also induced lipid peroxidation and altered the cellular viscoelasticity. The cytotoxicity, mitROS production, and decrease in the mitochondrial membrane potential with dabigatran etexilate were inhibited upon combined treatment with an antioxidant.

Dabigatran is an oral direct thrombin inhibitor. Idarucizumab reverses the effects of dabigatran by binding to it with an affinity that is 350 times higher than that of thrombin [22,23]. Dabigatran is an effective anticoagulant with a favorable bleeding profile in patients without regular monitoring. However, GI bleeding is reported as a side effect in patients treated with dabigatran [24,25]. Therefore, it is important to identify the mechanism underlying dabigatran-induced bleeding. Dabigatran etexilate and tartaric acid are the primary components of dabigatran. Dabigatran etexilate is absorbed at a low pH [26]; thus, a tartaric acid core is provided in dabigatran capsules [8]. Tartaric acid can create an acidic environment as it has pKa values of 2.98 and 4.34 [27]. Hence, it has been suggested that dabigatran cytotoxicity may be induced by the tartaric acid core contained within the capsule, which maintains the low-pH environment required to activate dabigatran etexilate. In this study, we compared the cytotoxicities of dabigatran etexilate and tartaric acid. Tartaric acid did not induce cellular damage at concentrations less than 10 mM (data not shown). The cell viability decreased by 40% after treatment with 100 μM dabigatran etexilate (Figure 1). However, the in vitro results may not necessarily indicate the in vivo results. At least in this study, the cytotoxicity of dabigatran etexilate was higher than that of tartaric acid. It is thought that dabigatran cytotoxicity may be induced by unconverted dabigatran etexilate. To our knowledge, this is the first study to investigate this hypothesis.

We evaluated the relationship between dabigatran and mitROS as a cytotoxic mechanism. We have previously reported the relationship between gastric mucosal injury and clinical drugs. Nonsteroidal anti-inflammatory drugs (NSAIDs) cause GI complications such as ulcers and erosions. This is most likely due to cyclooxygenase inhibition, prostaglandin deficiency, and neutrophil activation and infiltration, among other factors [28]. NSAIDs have also been reported to induce mitochondrial injury by reducing the mitochondrial transmembrane potential (MTP) [29]. We reported that indomethacin induces both MTP reduction and cellular apoptosis and that rebamipide exerts a protective effect on mitochondrial membrane stability [30]. We further reported that bisphosphonate, which is used for the treatment of postmenopausal osteoporosis, also induces injury via mitROS production, which can be prevented by antioxidants [14,15]. Based on these results, we concluded that the cytotoxicity of dabigatran etexilate may be strongly influenced by mitROS. Indeed, the intracellular fluorescence intensities of MitoSOX and DPPP were increased upon treatment with dabigatran etexilate (Figure 2 and Figure 3). Therefore, dabigatran etexilate induced mitROS production and membrane lipid peroxidation.

Membrane lipids are composed of one polar head and two hydrophobic hydrocarbon tails. Each tail is composed of saturated and poly-unsaturated fatty acyl chains. Lipid peroxidation is induced by a ROS-mediated attack on lipids containing carbon–carbon double bonds [31]. The fluidity of a cell membrane depends on its composition [32]. The length and saturation of fatty acid tails significantly affect the cell membrane fluidity. We have previously reported that promoting lipid peroxidation of cellular membranes alters the fluidity [33]. To evaluate the cell membrane fluidity, we measured the cellular membrane viscoelasticity using AFM. For a fluid mosaic model, the lateral movements and distributions of membrane components are restrained by such factors as the membrane domains, cytoskeletal interactions, extracellular matrix interactions, membrane protein interactions, and lipid–lipid interactions [34]. These membrane components determine the physical properties of the cells, such as viscoelasticity. Therefore, by measuring the viscoelasticity of cells, a correlation with membrane fluidity can be obtained. However, the methods for measuring the membrane fluidity without affecting the properties of the cell membrane are limited. Atomic force microscopy (AFM) has shown many successes in measuring the shape and mechanical properties of cells without affecting the cell membrane, such as the state of cytoskeletons, the elastic cytoplasm, and the viscoelasticity of cells [19,35,36,37,38]. In this study, the viscoelastic modulus of cells was obtained using the loss-tangent method, which is one of the AFM measurement methods. The viscoelastic loss tangent tan δ of a material is a dimensionless parameter that measures the ratio of energy dissipated to energy stored in one cycle of a periodic deformation, and it is the parameter for the viscoelasticity of a cell.

In this study, we focused on the relationship between mitROS production and cellular membrane viscoelasticity using AFM. Dabigatran etexilate can enhance mtROS production (Figure 2). An increase in mitROS production accelerates the changes in cell membrane fluidity and makes the stripe structure more elastic than the cell membrane, as shown in Figure 4C,D. The stripe structure is composed of actin filaments. As shown in Figure 4D, the number of actin filaments increased after dabigatran treatment. Since actin filaments are one of the factors that determine the elastic moduli of cells [39], their reduction reduces the elastic moduli. The values of tan δ decreased after dabigatran treatment, indicating that the viscosity of cells increased with dabigatran treatment. In particular, dabigatran treatment reduced the elastic modulus and increased the viscosity to change the viscoelasticity of the cells. Moreover, dabigatran etexilate is involved in cellular lipid peroxidation. Lipid peroxidation induces structural transitions in membranes and affects their orientation and fluidity [40]. Dabigatran etexilate altered the cellular viscoelasticity, actin formation, and lipid peroxidation. The lipid peroxidation in cellular membrane is promoted by ROS production. In this study, dabigatran etexilate increased the mitROS production. Taken together, dabigatran etexilate altered the properties of the cell membrane. Dabigatran etexilate also enhanced the mitROS production; thus, we considered that the alteration of the properties of the cell membrane may have been induced by increasing mitROS production.

We demonstrated that ascorbic acid could attenuate the cytotoxicity of dabigatran etexilate. Ascorbic acid is a typical antioxidant and it was found to prevent cellular injury and mitROS production induced by dabigatran etexilate (Figure 5 and Figure 6). The mitochondrial membrane potential was evaluated using JC-1, which is a cationic dye that accumulates in energized mitochondria. In cells with low MTP, JC-1 remains in a monomeric form that emits green fluorescence. In healthy cells with high MTP concentrations, JC-1 forms complexes known as J-aggregates with intense red fluorescence. A higher ratio of red to green fluorescence indicates greater mitochondrial membrane polarization [41]. Ascorbic acid can prevent the decrease in the mitochondrial membrane potential by dabigatran etexilate (Figure 7). It has been reported that the decrease in the mitochondrial membrane potential is the phenomena associated with apoptosis [42,43]. Ascorbic acid not only prevents mitochondrial fission but also decreases oxidative stress and apoptosis [44]. Therefore, increased mitROS production can accelerate cytotoxicity caused by dabigatran etexilate, especially in its contribution to apoptosis signaling. Ascorbic acid inhibits mitROS production and thereby prevents dabigatran etexilate cytotoxicity.

## 5. Conclusions

Dabigatran etexilate exhibited greater cytotoxicity than tartaric acid by inducing mitROS production. Therefore, the lowering of mitROS production by antioxidants, such as ascorbic acid, can help attenuate the cytotoxicity of dabigatran etexilate. We confirmed that dabigatran etexilate was responsible for dabigatran-mediated injury and concluded that dabigatran cytotoxicity can be inhibited by treatment with the antioxidant ascorbic acid.

## Figures and Tables

**Figure 1 cells-10-02508-f001:**
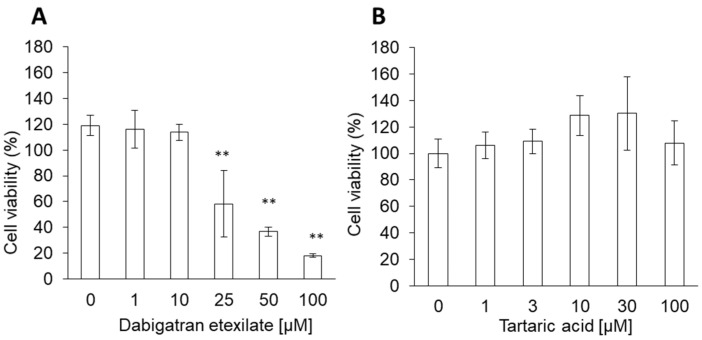
The cytotoxicity effect of (**A**) dabigatran etexilate and (**B**) tartaric acid. RGM1 cells were exposed to culture medium containing several concentrations of dabigatran etexilate or tartaric acid for 24 h, then a cell viability assay was performed. Data are expressed as means ± SD (n = 6). ** *p* < 0.01.

**Figure 2 cells-10-02508-f002:**
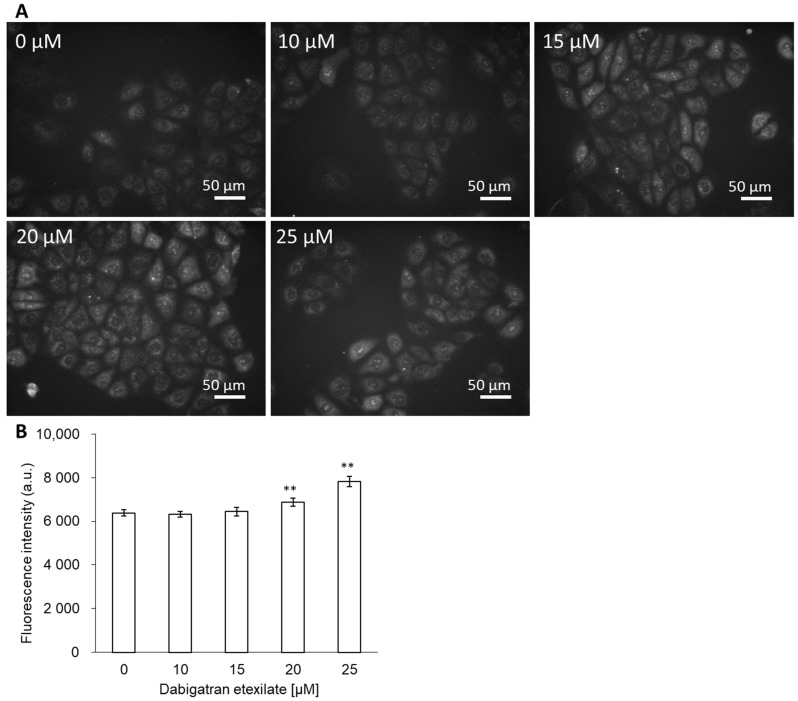
The fluorescence intensity of MitoSOX. (**A**) Fluorescent microscopy utilized to assess cellular uptake of MitoSOX. (**B**) Data are expressed as means ± SD (n =20). The sample size is the number of cells in the analyzed images. Ex = 535–555 nm and Em = 570–625 nm. ** *p* < 0.01.

**Figure 3 cells-10-02508-f003:**
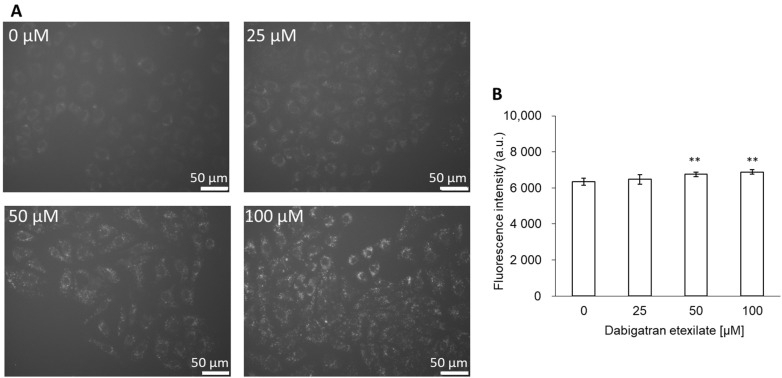
The fluorescence intensity of DPPP. (**A**) Fluorescent microscopy utilized to assess cellular uptake of DPPP. (**B**) Data are expressed as means ± SD (n = 10). The sample size is the number of cells in the analyzed images. Ex = 340–390 nm and Em = 420 nm. ** *p* < 0.01.

**Figure 4 cells-10-02508-f004:**
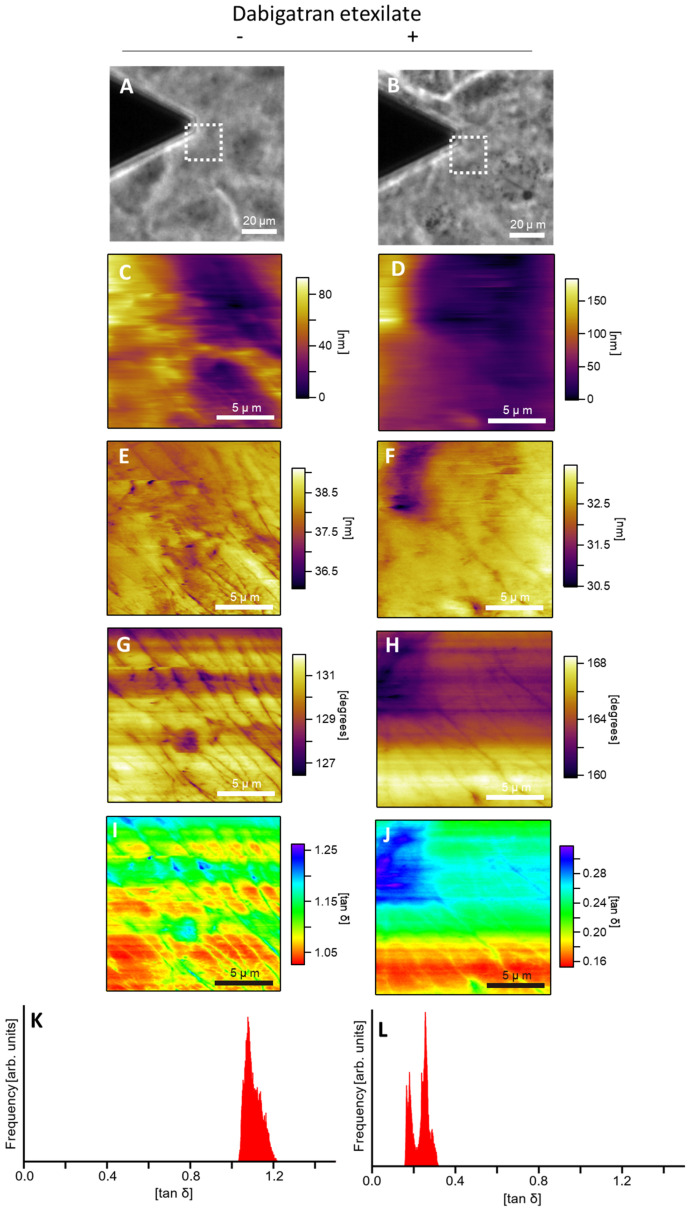
The cellular membrane viscosity. Cells were treated without (upper) and with (lower) dabigatran etexilate. (**A**,**B**) Phase contrast images; (**C**,**D**) AFM topography images; (**E**,**F**) AFM amplitude images; (**G**,**H**) AFM phase images; (**I**,**J**) loss tangent images; (**K**,**L**) histograms of the values of the loss tangent. The dashed lines in (**A**,**B**) indicate the AFM measurement region.

**Figure 5 cells-10-02508-f005:**
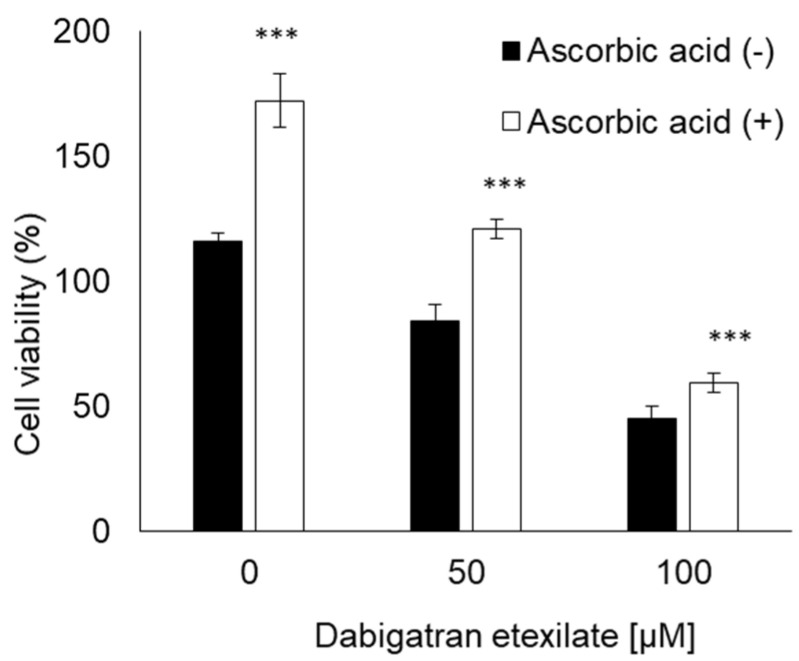
The cytotoxicity effect of dabigatran etexilate. Ascorbic acid prevented the dabigatran etexilate-induced cell injury. Data are expressed as means ± SD (n = 6). *** *p* < 0.001.

**Figure 6 cells-10-02508-f006:**
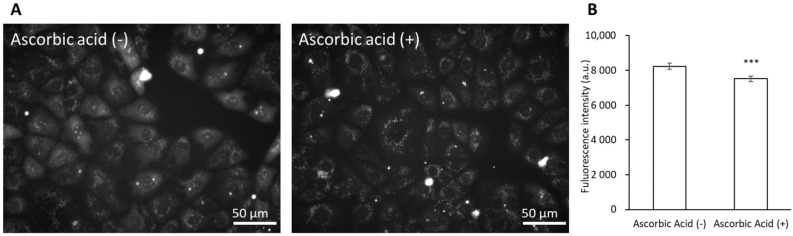
The fluorescence intensity of MitoSOX. The cells were incubated in a medium containing 25 μM dabigatran etexilate with or without 500 μM ascorbic acid. The cells were incubated for 6 h, the supernatant was aspirated, and MitoSOX was added. The fluorescence intensity of MitoSOX was measured using fluorescence microscopy. (**A**) Fluorescent microscopy utilized to assess cellular uptake of MitoSOX. (**B**) Data are expressed as means ± SD (n = 20). The sample size is the number of cells in the analyzed images. Ex = 535–555 nm and Em = 570–625 nm. *** *p* < 0.001.

**Figure 7 cells-10-02508-f007:**
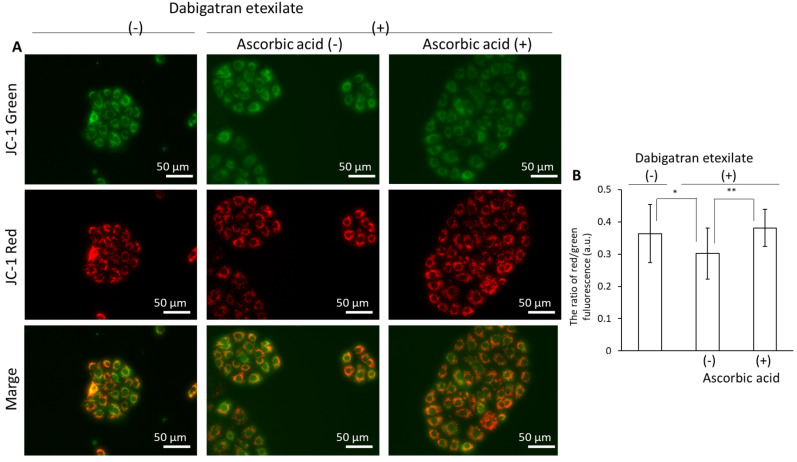
The fluorescence intensity of JC-1. The cells were incubated in a medium containing 25 μM dabigatran etexilate with or without 500 μM ascorbic acid for 6 h. After incubation, the supernatant was aspirated and the cells were incubated in a medium containing 2 μM JC-1. The fluorescence intensity of JC-1 was measured using fluorescence microscopy. (**A**) Fluorescent microscopy utilized to assess cellular uptake of JC-1. (**B**) Data are expressed as means ± SD (n = 20). The sample size is the number of cells in the analyzed images. * *p* < 0.05, ** *p* < 0.01.

## Data Availability

Not applicable.

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
