# Peer review of "Dabigatran Etexilate Induces Cytotoxicity in Rat Gastric Epithelial Cell Line via Mitochondrial Reactive Oxygen Species Production"

_cells, 2021, doi:10.3390/cells10102508_

Round 1

Reviewer 1 Report

The authors describe cytotoxic effects of dabigatran which they attribute to elevation of mitochondrial ROS production.

I have the following comments:

  1. The methodical details of cell viability determinations performed with a ‚cell counting kit-8‘ are not well described. What means WST (legend to Fig. 1)?
  2. Similarly the details of determinations of membrane fluidity are not clearly described. Since AFM measurements are not a routine approach for this much more details should be given.
  3. What means tan      ?
  4. It is not clear how the membrane fluidity and mitochondrial ROS might be related. That should be discussed in greater detail.
  5. The inhibitory effect of ascorbate is interesting, but rather unclear, since it should protect against ROS attacks in hydrophilic environment, but elevated lipid peroxidation (happening in a hydrophobic environment) was observed. That should be clarified. 

Author Response

  1. The methodical details of cell viability determinations performed with a cell counting kit-8‘ are not well described. What means WST (legend to Fig. 1)?

Cell Counting Kit-8 is a kit for measuring the number of cells in cell proliferation or chemical sensitivity tests. WST-8 ([2-(2-methoxy-4-nitrophenyl)-3-(4-nitrophenyl)-5-(2,4disulfophenyl)-2H-tetrazolium, monosodium salt]), which is a reagent, is reduced by intracellular dehydrogenase to produce water-soluble formazan, and the absorbance of this formazan at 450 nm can be directly measured to easily determine the number of viable cells.

We revised sentences following;

/ (2.2 Cell viability assay) The cell viability was evaluated using the Cell Counting Kit-8 (Dojindo, Tokyo, Japan) according to manufacturer’s protocol. Cell Counting Kit-8 is a kit for measuring the number of cells in cell proliferation or chemical sensitivity tests. WST-8 ([2-(2-methoxy-4-nitrophenyl)-3-(4-nitrophenyl)-5-(2,4disulfophenyl)-2H-tetrazolium, monosodium salt]), which is a reagent, is reduced by intracellular dehydrogenase to produce water-soluble formazan, and the absorbance of this formazan at 450 nm can be directly measured to easily determine the number of viable cells.

/ (Legend in Fig. 1) then cell viability assay was performed.

  1. Similarly the details of determinations of membrane fluidity are not clearly described. Since AFM measurements are not a routine approach for this much more details should be given.

According to reviewer’s comment, we added the sentence in discussion;

For fluid mosaic model, the lateral movements and distributions of membrane components are restrained by such as, the membrane domains, cytoskeletal interactions, extracellular matrix interactions, membrane protein interactions, lipid–lipid interactions[35]. These membrane components determine the physical properties of the cells, such as viscoelastic. Therefore, measuring the viscoelasticity of cells, a correlation with membrane fluidity can be obtained. However, there are limited methods for measuring the membrane fluidity without affecting the properties of the cell membrane. Atomic force microscope (AFM) has had many successes measuring extremely the shape and mechanical properties of cells without affecting the cell membrane, such as the state of cytoskeletons, the elastic cytoplasm, and viscoelasticity of cells[20,36-39]. In this study, the viscoelastic modulus of cells was obtained using the lost tangent method, which is one of the AFM measurement methods. The viscoelastic loss tangent tan d of a material is a dimensionless parameter that measures the ratio of energy dissipated to energy stored in one cycle of a periodic deformation, and is the parameter for the viscoelasticity of the cell.

  1. What means tan?

[tan d] is the value of loss tangent. The value of loss tangent is estimated by following eq.[6]

Here, w, Q, V, and j are the cantilever drive frequency, quality factor, the cantilever amplitude, and the phase, respectively. wfree and Vfree are the free reference values of the drive frequency and the amplitude when the cantilever is separated from the sample by a height of Dz.

We added this sentence in (2.5. Measurement of cell membrane viscosity using atomic force microscopy).

  1. It is not clear how the membrane fluidity and mitochondrial ROS might be related. That should be discussed in greater detail.

We revised sentences following in (Discussion);

In this study, we focused the relationship between the mitROS production and the cellular membrane viscoelasticity using AFM. Dabigatran etexilate can enhance the mtROS production (Figure 2). An increase in mitROS production accelerates the changes in cell membrane fluidity and makes the stripe structure more elastic than the cell membrane, as shown in Figures 4C and 4D. The stripe structure is composed of actin filaments. As shown in Figure 4D, the number of actin filaments increased after dabigatran treatment. Since actin filaments are one of the factors that determine the elastic moduli of cells [40], their reduction reduces the elastic moduli. The values of tan  d decreased after dabigatran treatment, indicating that the viscosity of cells increased with dabigatran treatment. In particular, dabigatran treatment reduced the elastic modulus and increased the viscosity to change the viscoelasticity of the cells. Moreover, dabigatran etexilate is involved in cellular lipid peroxidation. Lipid peroxidation induces structural transitions in membranes and affects their orientation and fluidity [41]. Dabigatran etexilate altered the cellular viscoelasticity, actin formation, and lipid peroxidation. The lipid peroxidation in cellular membrane is promoted by ROS production. In this study, dabigatran etexilate icrease the mitROS production. Taken together, dabigatran etexilate altered the properties of the cell membrane. Dabigatran etexilate also enhance the mitROS production, thus we considered that the alteration of the properties of the cell membrane may be induced by increasing mitROS production.

  1. The inhibitory effect of ascorbate is interesting, but rather unclear, since it should protect against ROS attacks in hydrophilic environment, but elevated lipid peroxidation (happening in a hydrophobic environment) was observed. That should be clarified.

We can not demonstrate the inhibition effect of lipid peroxidation in ascorbic acid. However, ascorbic acid can inhibit the cytotoxicity by dabigatran (Fig. 5) and the production of mitROS (Fig. 6). Moreover, Ascorbic acid can inhibit the cell apoptosis derived dabigatran (Fig. 7). From these results, we considered that ascorbic acid can inhibit the lipid peroxidation.

Reviewer 2 Report

The manuscript outlines a study to assess the cause of cytotoxicity associated with dabigatran by assessing the effect of dabigatran etexilate and tartaric acid  treatment on rat gastric epithelial cells.

This is an interesting study, but I have some comments/questions for the authors:

  1. The introduction doesn`t really make it clear why mitochondrial ROS production should be investigated as a cause of the gastric bleeding associated with dabigatran treatment. I think the authors should add more information to justify why they  primarily selected  mitochondrial ROS production as a cause of the cytotoxicity associated with dabigatran treatment and not other cellular dysfunction.
  2. Some justification on the concentrations of dabigatran etexilate, tartaric acid and ascorbate acid  used in this study should be provided as the concentrations just appear without explanation in the methods section.
  3. Why were only 6 hour incubation periods with the different compounds used?
  4. Did dabigatran etexilate or tartaric acid have any interaction with the lipid peroxidation, viscosity or JC-1 assays used in this study? No information provided.
  5. Was the decreased  elasticity and increased the viscosity of the cell following Dabigatran etexilate treatment caused directly by a chemical interaction of the prodrug with the cell membrane or was it due to increased ROS generation?
  6. Can the authors be confident that all the  cytotoxicity  associated with Dabigatran is in fact mainly due to its  prodrug?
  7. A clearer explanation and possibly a schematic diagram would be appropriate in the discussion section to explain the mechanism by which Dabigatran etexilate induces an increased in mitochondrial ROS and consequently this may then lead to cellular toxicity and consequently gastric bleeding.
  8. Why did the authors only use ascorbate acid as the antioxidant of choice and would other antioxidants be expected to elicit a similar response?
  9. I think the title of the paper should be changed as there is no assessment of gastric injury per se only  a loss of viability of immortalise rat epithelial cells.

Author Response

Reviwer2

  1. The introduction doesn`t really make it clear why mitochondrial ROS production should be investigated as a cause of the gastric bleeding associated with dabigatran treatment. I think the authors should add more information to justify why they primarily selected mitochondrial ROS production as a cause of the cytotoxicity associated with dabigatran treatment and not other cellular dysfunction.

According to reviewer’s comment, we added the sentence in introduction.

Because indometacine and bisphosphonate induce GI bleeding and cytotoxicity via mi-tROS production, we focused the mitROS.

  1. Some justification on the concentrations of dabigatran etexilate, tartaric acid and ascorbate acid  used in this study should be provided as the concentrations just appear without explanation in the methods section.

The concentration was set to confirm the concentration at which a difference in cytotoxicity occurs between dabigatran etexilate and tartaric acid. According to reviewer’s comment, we added the sentence in materials and methods.

We have confirmed that this ascorbic acid concentration did not cell injury in RGM1 (data not shown).

  1. Why were only 6 hour incubation periods with the different compounds used?

The cell viability in 25 µM dabigatran etexilate was decreased until 50% for 24 h incubation (Fig. 1A). In this study, to evaluate the mechanisms of cell injury by dabigatran etexilate, we select the 0 to 25 μM concentration and 6 h incubation time.

We added this sentence in 3.2. Dabigatran increases mitROS production

  1. Did dabigatran etexilate or tartaric acid have any interaction with the lipid peroxidation, viscosity or JC-1 assays used in this study? No information provided.

We evaluated the relationship between dabitgatran etexilate and lipido peroxidation, viscosity or JC-1 assay. We added the sentence in Discussion;

The cytotoxicity, mitROS production and decrease of the mitochondrial membrane potential with dabigatran etexilate were inhibited upon combined treatment with an antioxidant.

  1. Was the decreased elasticity and increased the viscosity of the cell following Dabigatran etexilate treatment caused directly by a chemical interaction of the prodrug with the cell membrane or was it due to increased ROS generation?

We have previously reported that promoting lipid peroxidation of cellular membranes alters the fluidity. The lipid peroxidation in cellular membrane is promoted by ROS production. In this study, dabigatran increased ROS production and lipid peroxidation of membranes. We considered that the changes in viscosity and elasticity due to increase of ROS production.

We revised the sentence in Discussion;

Dabigatran etexilate altered the cellular viscoelasticity, actin formation, and lipid peroxidation. The lipid peroxidation in cellular membrane is promoted by ROS production. In this study, dabigatran etexilate icrease the mitROS production. Taken together,...

  1. Can the authors be confident that all the cytotoxicity associated with Dabigatran is in fact mainly due to its prodrug?

The cell viability was decreased until 20% in 100 µM dabigatran etexilate, while it did not decrease in 100 µM tartaric acid. At least, we considered that cell injury of dabigatran was induced by dabigatran etexilate in our results.

  1. A clearer explanation and possibly a schematic diagram would be appropriate in the discussion section to explain the mechanism by which Dabigatran etexilate induces an increased in mitochondrial ROS and consequently this may then lead to cellular toxicity and consequently gastric bleeding.

We revised the sentence in Discussion;

In this study, we focused the relationship between the mitROS production and the cellular membrane viscoelasticity using AFM. Dabigatran etexilate can enhance the mtROS production (Figure 2). An increase in mitROS production accelerates the changes in cell membrane fluidity and makes the stripe structure more elastic than the cell membrane, as shown in Figures 4C and 4D. The stripe structure is composed of actin filaments. As shown in Figure 4D, the number of actin filaments increased after dabigatran treatment. Since actin filaments are one of the factors that determine the elastic moduli of cells [40], their reduction reduces the elastic moduli. The values of tan d decreased after dabigatran treatment, indicating that the viscosity of cells increased with dabigatran treatment. In particular, dabigatran treatment reduced the elastic modulus and increased the viscosity to change the viscoelasticity of the cells. Moreover, dabigatran etexilate is involved in cellular lipid peroxidation. Lipid peroxidation induces structural transitions in membranes and affects their orientation and fluidity [41]. Dabigatran etexilate altered the cellular viscoelasticity, actin formation, and lipid peroxidation. The lipid peroxidation in cellular membrane is promoted by ROS production. In this study, dabigatran etexilate icrease the mitROS production. Taken together, dabigatran etexilate altered the properties of the cell membrane. Dabigatran etexilate also enhance the mitROS production, thus we considered that the alteration of the properties of the cell membrane may be induced by increasing mitROS production.

We demonstrated that ascorbic acid could attenuate the cytotoxicity of dabigatran etexilate. Ascorbic acid is a typical antioxidant and prevented cellular injury and mitROS production induced by dabigatran etexilate (Figures 5 and 6). The mitochondrial membrane potential was evaluated using JC-1, which is a cationic dye that accumulates in energized mitochondria. In apoptotic cells with low MTP, JC-1 remains in a monomeric form that emits green fluorescence. In healthy cells with high MTP concentrations, JC-1 forms complexes known as J-aggregates with intense red fluorescence. A higher ratio of red to green fluorescence indicates greater mitochondrial membrane polarization [42]. Ascorbic acid can prevent the decrease of the mitochondrial membrane potential by dabigatran etexilate (Figure 7). It is reported that the decrease of the mitochondrial membrane potential is the phenomena associated with apoptosis [43,44]. Ascorbic acid not only prevents mitochondrial fission but also decreases oxidative stress and apoptosis [45]. Therefore, increased mitROS production can accelerate cytotoxicity caused by dabigatran etexilate especially in contibute to apoptosis signaling. Ascorbic acid inhibits mitROS production and thereby prevents dabigatran etexilate cytotoxicity.

  1. Why did the authors only use ascorbate acid as the antioxidant of choice and would other antioxidants be expected to elicit a similar response?

In this study, we select the ascorbic acid because it is a typical antioxidant. We consider that the antioxidant, which scavenge radical directly like ascorbic acid, may be attenuate the cytotoxicity by dabigatran etexilate.

We added the sentence in discussion;

Ascorbic acid is a typical antioxidant and prevented cellular injury and mitROS production induced by dabigatran etexilate (Figures 5 and 6).

  1. I think the title of the paper should be changed as there is no assessment of gastric injury per se only a loss of viability of immortalise rat epithelial cells.

According to reviewer’s comment, we revised title to “Dabigatran etexilate induces cytotoxicity in rat gastric epithelial cell line via mitochondrial reactive oxygen species production”

Reviewer 3 Report

The manuscript addresses an important issue of the mechanisms of cytotoxicity of dabigatran, the anticoagulant drug. The study is well performed and described, however the following aspects shall be improved:

  1. The conclusions on the causal relationships between the measured parameters shall be drawn more carefully. For example, in lines 296-297:, „dabigatran etexilate altered the properties of the cell membrane by increasing ROS production”. The presented results show only the co-occurrence of both events. To conclude that changes in membrane properties are downstream of the increase in mitoROS, additional controls are needed, for example checking the effect of ascorbate on cell membrane parameters in dabigatran etexilate-treated cells (as it was done in the case of cell viability test) as well as comparing the effects of dabigatran etexilate with the effects of other agents increasing mitoROS in the tested cells.
  2. Using JC-1 as an indicator of apoptosis is not correct. It measures mitochondrial membrane depolarization, which may ocurr under many conditions besides apoptosis. Without checking some other apoptosis markers, no conclusions can be drawn concerning the presence of apoptosis in the described model. However, the results of mitochondrial mambrane potential give valuable information on the condition and physiological state of mitochondria, which is important to clarify the reasons of the elevation in mitROS. Thus, the measuremens of JC-1 fluorescence shall be discussed in this context, not as apoptosis markers.
  3. The legends to figures 6 and 7 are a bit confusing. If I understand well, thet they present the effects of ascorbic acid in dabigatran etexilate-treated cells but it is not completely clear. If so, the controls without both dabigatran etexilate and ascorbic acid shall also be shown, to see the scale of the reversal of unfavorable effects – particularly in case of JC-1 measurements, as the information on how dabigatran etexilate alone affects mitochondrial membrane potential is missing from the manuscript.
  4. The information on the number of independent experimental repetitions is missing. N=20 is given in the figure legends but it is not explained whether it is the number of independent repetitions (from indicidual cell cultures) or the numer of the analyzed images or cells.

Author Response

Reviwer3

  1. The conclusions on the causal relationships between the measured parameters shall be drawn more carefully. For example, in lines 296-297:, „dabigatran etexilate altered the properties of the cell membrane by increasing ROS production”. The presented results show only the co-occurrence of both events. To conclude that changes in membrane properties are downstream of the increase in mitoROS, additional controls are needed, for example checking the effect of ascorbate on cell membrane parameters in dabigatran etexilate-treated cells (as it was done in the case of cell viability test) as well as comparing the effects of dabigatran etexilate with the effects of other agents increasing mitoROS in the tested cells.

In previous study, we reported that indomethacin, which induce the ROS production and lipid peroxidation, alter the cellular membrane fluidity. In this study, we cleared that mitROS can alter the membrane viscoelasticity. Now we evaluate the antioxidant effect to alter the cellular membrane viscoelasticity by dabigatran using antioxidant derived algae. We will report between alteration of the cellular membrane viscoelasticity by dabigatran and antioxidant in next paper.

According to reviewer’s comment, we revised the sentence in discussion;

Taken together, dabigatran etexilate altered the properties of the cell membrane. Dabigatran etexilate also enhance the mitROS production, thus we considered that the al-teration of the properties of the cell membrane may be induced by increasing mitROS production.

  1. Using JC-1 as an indicator of apoptosis is not correct. It measures mitochondrial membrane depolarization, which may ocurr under many conditions besides apoptosis. Without checking some other apoptosis markers, no conclusions can be drawn concerning the presence of apoptosis in the described model. However, the results of mitochondrial mambrane potential give valuable information on the condition and physiological state of mitochondria, which is important to clarify the reasons of the elevation in mitROS. Thus, the measuremens of JC-1 fluorescence shall be discussed in this context, not as apoptosis markers.

According to reviewer’s comment, we revised the sentence in 3.5 Ascorbic acid suppresses the cytotoxicity of dabigatran etexilate via the inhibition of mi-tROS production;

The cells were incubated with ascorbic acid to suppress the cytotoxicity of dabigatran etexilate. Compared to untreated cells, the ascorbic acid-treated cells remained viable (Figure 5). Ascorbic acid can inhibit ROS generation induced by dabigatran etexilate, as measured using MitoSOX. Dabigatran etexilate induced intracellular ROS production, which was subsequently inhibited in response to treatment with ascorbic acid (Figures 6A and 6B). Moreover, the mitochondrial membrane potential was observed using JC-1 under a fluorescence microscope. JC-1 green fluorescence in the dabigatran etexilate treated cells without ascorbic acid was greater than that in the cells with ascorbic acid, suggesting that dabigatran etexilate-induced the decrease of the mitochondrial membrane potential apoptosis was suppressed by ascorbic acid (Figure 7).

And we revised the sentence in discussion;

A higher ratio of red to green fluorescence indicates greater mitochondrial membrane polarization [37]. Ascorbic acid can prevent the decrease of the mitochondrial membrane potential by dabigatran etexilate (Figure 7). It is reported that the decrease of the mitochondrial membrane potential is the phenomena associated with apoptosis. Ascorbic acid not only prevents mitochondrial fission but also decreases oxidative stress and apoptosis [38]. Therefore, increased mitROS production can accelerate cytotoxicity caused by dabigatran etexilate especially in contibute to apoptosis signaling. Ascorbic acid inhibits mitROS production and thereby prevents dabigatran etexilate cytotoxicity.

  1. The legends to figures 6 and 7 are a bit confusing. If I understand well, thet they present the effects of ascorbic acid in dabigatran etexilate-treated cells but it is not completely clear. If so, the controls without both dabigatran etexilate and ascorbic acid shall also be shown, to see the scale of the reversal of unfavorable effects – particularly in case of JC-1 measurements, as the information on how dabigatran etexilate alone affects mitochondrial membrane potential is missing from the manuscript.

According to reviewer’s comment, we revised the sentence in 3.5 Ascorbic acid suppresses the cytotoxicity of dabigatran etexilate via the inhibition of mi-tROS production;

The cells were incubated with ascorbic acid to suppress the cytotoxicity of dabigatran etexilate. Compared to untreated cells, the ascorbic acid-treated cells remained viable (Figure 5). Ascorbic acid can inhibit ROS generation induced by dabigatran etexilate, as measured using MitoSOX. Dabigatran etexilate induced intracellular ROS production, which was subsequently inhibited in response to treatment with ascorbic acid (Figures 6A and 6B). Moreover, the mitochondrial membrane potential was observed using JC-1 under a fluorescence microscope. JC-1 green fluorescence in the dabigatran etexilate treated cells without ascorbic acid was greater than that in the cells with ascorbic acid, suggesting that dabigatran etexilate-induced the decrease of the mitochondrial membrane potential apoptosis was suppressed by ascorbic acid (Figure 7).

And we revised the Figure 7.

  1. The information on the number of independent experimental repetitions is missing. N=20 is given in the figure legends but it is not explained whether it is the number of independent repetitions (from indicidual cell cultures) or the numer of the analyzed images or cells.

N=20 is the number of the analyzed images or cells. We added the sentence in legends (Fig. 2, Fig. 3, Fig. 6 and Fig. 7).

Round 2

Reviewer 1 Report

The authors addressed my comments and revised their manuscript accordingly. There are still some minor problems with the language which should be corrected. E.g. lines 316-317:

'These membrane components determine the physical properties of the cells, such as viscoelasticity. (...ity missing)'

Author Response

The authors addressed my comments and revised their manuscript accordingly. There are still some minor problems with the language which should be corrected. E.g. lines 316-317:

'These membrane components determine the physical properties of the cells, such as viscoelasticity. (...ity missing)'

> According to reviewer’s comment, we revised this sentence.

Reviewer 3 Report

Some aspects have been improved, but other points still need correction:

  1. Explanation that "n is the number of the analyzed images or cells" (in figure legends) is unclear and not precise. Moreover, the information on the number of independent biological replicates is still mising. Good practice is to repeat the experiment at least three times independently (independent cell seeding, treatment, probe loading and measurements) to confirm the effect.
  2. The formuation "the decrease of the mitochondrial membrane potential apoptosis was suppressed..." (line 246) is unclear.
  3. Line 351: "In apoptotic cells with low MTP, JC-1 remains in a monomeric form" - as mentioned before, apoptosis is onle one out of the many conditions when a decrease of MTP can be observe, thus this sentence is misleading. MTP measuremets shall not be discussed in the context of the presence of apoptosis if no apoptosis marker has been determined in the presented study.
  4. JC-1 experiments still miss the appropriate reference values. Mitochondrial membrane potential has not been determined in untreated cells to compare it with dabigatran-treated cells. Therefore it cannot be stated that "Ascorbic acid can prevent the decrease of the mitochondrial membrane potential by dabigatran etexilate" (line 354), as the depolarizing effect of dabigatran has not been shown (it is likely but has not been checked).
  5. The legends to figures 6 and 7 have not been corrected to clarify what treatments exactly have been applied.

Author Response

1. Explanation that "n is the number of the analyzed images or cells" (in figure legends) is unclear and not precise. Moreover, the information on the number of independent biological replicates is still mising. Good practice is to repeat the experiment at least three times independently (independent cell seeding, treatment, probe loading and measurements) to confirm the effect.

>According to reviewer’s comment, we revised the legend following;

Sample size is the number of cells in the analyzed images.

>And added the sentence in 2.7. Statistical analysis;

Each experiment was performed independently at least three times.

2. The formuation "the decrease of the mitochondrial membrane potential apoptosis was suppressed..." (line 246) is unclear.

> According to reviewer’s comment, we revised the sentence; “the decrease of the mitochondrial membrane potential was suppressed by ascorbic acid (Figure 7).”

3. Line 351: "In apoptotic cells with low MTP, JC-1 remains in a monomeric form" - as mentioned before, apoptosis is onle one out of the many conditions when a decrease of MTP can be observe, thus this sentence is misleading. MTP measuremets shall not be discussed in the context of the presence of apoptosis if no apoptosis marker has been determined in the presented study.

> According to reviewer’s comment, we revised the sentence following;

The cells with low MTP, JC-1 remains in a monomeric form that emits green fluorescence.

4. JC-1 experiments still miss the appropriate reference values. Mitochondrial membrane potential has not been determined in untreated cells to compare it with dabigatran-treated cells. Therefore it cannot be stated that "Ascorbic acid can prevent the decrease of the mitochondrial membrane potential by dabigatran etexilate" (line 354), as the depolarizing effect of dabigatran has not been shown (it is likely but has not been checked).

> According to reviewer’s comment, we revised the data of untreated cells. >Compared to dabigatran etexilate treatment cells, the mitochondrial membrane potential in untreated cells wes higher.

>We revised the sentence in 3.5 Ascorbic acid suppresses the cytotoxicity of dabigatran etexilate via the inhibition of mi-tROS production;

Moreover, the mitochondrial membrane potential was observed using JC-1 under a fluorescence microscope. JC-1 green fluorescence in the dabigatran etexilate treated cells was lower than untreated cells. Moreover, JC-1 green fluorescence in the dabigatran etexilate treated cells without ascorbic acid was greater than that in the cells with ascorbic acid, suggesting that dabigatran etexilate-induced the decrease of the mitochondrial membrane potential was suppressed by ascorbic acid (Figure 7).

5. The legends to figures 6 and 7 have not been corrected to clarify what treatments exactly have been applied.

> According to reviewer’s comment, we revised the sentence following;

Figure 6 The fluorescence intensity of MitoSOX. The cells were incubated in a medium containing 25 μM dabigatran etexilate with or without 500 μM ascorbic acid. The cells were incubated for 6 h, the supernatant was aspirated, and MitoSOX was added. The fluorescence intensity of mitoSOX was measured using fluorescence microscopy. (A) Fluorescent microscopy utilized to assess cellular uptake of MitoSOX. (B) Data are ex-pressed as means ± SD (n = 20). The number of n is the number of the analyzed images of cells. Ex = 535-555 nm and Em = 570-625 nm. *** p < 0.001.

Figure 7 The fluorescence intensity of JC-1. The cells were incubated in a medium medium containing 25 μM dabigatran etexilate with or without 500 μM ascorbic acid for 6 h. After incubation, the supernatant was aspirated and the cells were incubated in a medium containing 2 μM JC-1. The fluorescence intensity of mitoSOX was measured using fluorescence microscopy. (A) Fluorescent microscopy utilized to assess cellular uptake of JC-1. (B) Data are expressed as means ± SD (n = 20). The number of n is the number of the analyzed images of cells. *p < 0.05, **p < 0.01.

Round 3

Reviewer 3 Report

All the issues raised in the review have been addressed, however they still need some corrections due to certain slopiness of the authors while introducing the changes. I kindly request the authors to do it carefully, to avoid further review rounds.

In the lines 245-249, the authors added the sentence "JC-1 green fluorescence in the dabigatran etexilate treated cells was lower than untreated cells", which suggests an increase in mitochondrial membrane potential upon dabigatran etexilate treatment. It does not fit to the further statement about "dabigatran etexilate-induced decrease of the mitochondrial membrane potential". Probably the authors ment lower red/green ratio, not "green fluorescence", but this has to be clarified. Also, in the plot in the figure 7, the axis is labeled "fluorescence intensity - a.u.", while in JC-1 measurements the "red/green fluorescence ratio" shall be presented. Additionally, in the legend "fluorescence intensity of mitoSOX" is mentioned instead of JC-1.

Line 360: shall be "In the cells with low MTP..." 

Author Response

According to reviewer’s comments, we revised the sentence;

Moreover, the mitochondrial membrane potential was observed using JC-1 under a fluorescence microscope. The ratio of red/ green fluorescence in the dabigatran etexilate treated cells was lower than that in the untreated cells. Moreover, this ratio in the dabigatran etexilate treated cells with ascorbic acid was greater than that in the cells without ascorbic acid, suggesting that dabigatran etexilate-induced the decrease of the mitochondrial membrane potential was suppressed by ascorbic acid (Figure 7).

According to reviewer’s comments, we revised figure 7 and legend of figure 7.

According to reviewer’s comments, we revised in the line 360.